# Construction of a public health emergency information system framework: A case study of Zhuhai city, China

**Sicheng Huang[1], Xuebao Zhang[1], Long Chen[1], Xihe Ni[1], Ying Fan[1], Chaomin Zhao[1], Junfeng Xiao[2], Feng Ruan [3]***

**1** Department of Emergency Management, Zhuhai Center for Disease Control and Prevention, Zhuhai, Guangdong, China, **2** Institute of Public Health and Research, Zhuhai Center for Disease Control and Prevention, Zhuhai, Guangdong, China, **3** Center Leadership Office, Zhuhai Center for Disease Control and Prevention, Zhuhai, Guangdong, China

* zhcdcyjb@163.com

## Abstract

### Background

A public health emergency information system serves as a critical tool for collecting and analyzing data from sudden public health events, thereby providing a scientific basis for governmental decision-making. However, research on the systematic construction of such information system frameworks within China's public health infrastructure is lacking.

### Objective

Taking Zhuhai city as a case study, this study aims to construct a comprehensive public health emergency information system framework applicable to public health departments at the municipal, county, and street/township levels.

### Methods

First, through a literature review and expert group discussion, the preliminary framework of system indicators is determined. Second, through two rounds of the Delphi method, 41 experts are invited to qualitatively select the system framework indicators, with the aim of obtaining consensus among experts. Finally, the system is improved through application, feedback, and redesign.

### Results

After two rounds of consultation, the final system at the city and county levels consists of 5 first-level indicator modules and 21 second-level indicator modules, whereas the system at the city, county, and street/township levels consists of 4 first-level indicator modules and 17 second-level indicator modules. Most of the indicators in the "emergency preparedness" and "emergency response" modules are

**Data availability statement:** All data are in the manuscript and/or Supporting information files.

**Funding:** This work was supported by Hygiene and Health Bureau Fund Project of Zhuhai (2220009000231).

**Competing interests:** The authors declare no competing interests.

considered important and should be retained as they can play a role in collecting and analysing information on infectious disease outbreaks through practical applications.

## Conclusion

The public health emergency information system constructed in this study can be applied to public health departments such as disease prevention and control centres. Promotion can improve the efficiency of handling infectious disease outbreaks and provide a scientific basis for decision-making analysis.

## Background

A public health emergency information system (PHEIS) is a technology and research method that uses epidemiology, statistics, and computer science to collect real-time and reliable data during public health emergencies, providing the best solution for epidemic control [1]. It is mainly divided into two parts; the first part consists of information data collection, cleaning, preprocessing, while the second part consists of management, and data analysis and interpretation [2]. Since the COVID-19 epidemic, the role of PHEISs has become increasingly prominent. When an epidemic breaks out, timely and accurate data collection is crucial for commanders to make decisions; however, it is often difficult to obtain such data. With the increasing nature and complexity of emergency management for infectious diseases, the processes of data collection, management, and reporting face challenges.

In recent years, research on information collection and management systems in the field of public health has focused mainly on chronic disease monitoring in hospitals [3] and community clinical practice [4]; this approach has certain limitations in regard to promoting these methods in the field of epidemic response. In addition, there have been studies that have focused on information collection and management systems for certain infectious disease outbreaks, such as the African Ebola epidemic information system [5], the application of Brazil's national health information system in collecting Zika epidemic information [2], and the application of Liberia's comprehensive disease surveillance system in collecting Ebola epidemic information [6]. However, these studies have focused mainly on a specific infectious disease and thus have limitations in their application to other infectious disease outbreaks and public health emergencies. In terms of research methods, most of the abovementioned studies have used literature reviews [2], expert experience, and interviews [6] to introduce the construction process of information systems, thus demonstrating a lack of scientific research methods. Alternatively, such studies have focused on the introduction of the construction process of mathematical statistical models [7] and computer programming fields [1,8], thereby demonstrating a lack of elaboration in the field of epidemiology. In summary, there is currently a lack of research on PHEISs in the field of infectious disease outbreaks.

At present, many studies on the construction of PHEISs, both domestically and internationally, have been based on official policy documents and relevant literature

for system framework construction [6]; most of these studies have focused on only national-, provincial-, and city-level system construction. In recent years, with the establishment of street/township-level disease control centres in many cities in China, frontline disposal work has gradually shifted to include district- and street-level public health departments. Therefore, it is necessary to build PHEISs at the city, district, and township/street levels so that the systems can achieve vertical commands from top to bottom in daily operation.

Zhuhai is a prefecture-level city located in Guangdong Province, China; it had a permanent population of 2.4941 million at the end of 2023. After the outbreak of COVID-19 in 2020, Zhuhai established five district centres for disease control and prevention. In 2024, Zhuhai city established 21 additional street/township-level disease control centres, and front-line disposal work began to gradually shift to district- and street-level disease control departments. Therefore, this study aims to simultaneously construct PHEISs at the Zhuhai city and district levels, as well as at the city, district, and street/township levels, to meet current needs.

Most of the research on the construction of PHEISs, both at home and abroad, uses official policy documents and relevant literature to construct systematic frameworks [3] that collaborate closely with stakeholders (government, frontline public health professionals, engineers) during the system design phase for joint development. However, this method has a strong level of subjectivity in the qualitative selection stage of the indicator framework. To address the shortcomings of this method, in this study, we combine this stage with the Delphi method for construction. The Delphi method can be applied to establish various evaluation index systems and determine specific indicators. Through several rounds of feedback, we fully utilize and absorb the experience and knowledge of the experts, gradually converging their opinions. The process of combining the expert experience method and the Delphi method is currently widely used in the construction of health information systems [9].

## Method

### Design

We aimed to combine the opinions and needs of numerous stakeholders to provide an optimal solution. In terms of participating in system design, stakeholders within the system (leaders, emergency management experts, and onsite disposal personnel of the city and district centres for disease control and prevention) were invited to conduct research and develop a preliminary framework, which includes components and a list of elements for system design [9].

### Ethics approval and informed consent

This study was approved by the Medical Research Ethics Committee of the Zhuhai Center for Disease Control and Prevention in Guangdong Province, China. All the research procedures complied with the ethical standards of the institutional research committee. Owing to the extremely low risk of this study and the absence of a routine written consent procedure, the Institutional Review Board (IRB) granted a waiver of written informed consent for the subjects. The study invitation letter detailed the purpose and process of the study, and experts' completion and submission of an online questionnaire is considered their voluntary informed consent to participate. This study was conducted in strict accordance with the Declaration of Helsinki.

### Modified e-Delphi approach

This study employed a modified e-Delphi approach. The design and reporting of the study were guided by the core principles of the Conducting and REporting DElphi Studies (CREDES) guidelines [10]. In contrast to the classical Delphi method—which typically commences with an open-ended qualitative round to generate an initial item pool—a modified Delphi method uses a structured questionnaire derived from a systematic literature review as the basis for the first round of consultation. All the consultation procedures were conducted electronically via email using Excel-based survey forms,

and no face-to-face meetings were held. Experts were permitted to reevaluate and revise their initial ratings within one week of submission. During this study, we invited 41 experts in the field of public health in the selected city to not only screen and revise indicators through two rounds of the Delphi method [11] but also determine indicators and weights.

## Expert eligibility criteria and sampling strategy

To ensure the professionalism and authority of the expert consultation and considering the public health emergency response context of this study, the following inclusion criteria were established: (1) Professional title and education: Experts were required to hold at least an intermediate professional title in a relevant field or possess a postgraduate degree or higher. (2) Years of experience: A minimum of two years of cumulative professional experience in public health, emergency management, or related fields was needed. (3) Role and responsibilities: Eligible participants included frontline response personnel and managerial staff from the emergency response departments of centers for disease control and prevention at various administrative levels. Individuals were excluded if they failed to meet any of the above criteria, did not respond to the invitation within the specified timeframe (21 days), or submitted an incomplete first-round questionnaire that remained incomplete after a reminder was sent.

A purposive sampling strategy was employed for expert selection. Through routine professional contacts and daily work interactions, the research team identified individuals who served as responders and managers for public health emergency events at various levels of centers for disease control and prevention in advance. These individuals were then evaluated against the predefined eligibility criteria, and those meeting all the inclusion requirements were purposefully selected and invited to participate as core experts.

## Expert recruitment process

First, we constructed a preliminary indicator framework. The data sources of this study included the framework of domestic and international infectious disease epidemic information collection systems, the framework of Guangdong Province's health emergency information system, government documents such as the emergency response plan and technical scheme for public health emergencies in Zhuhai city, and national health emergency work standards for disease prevention and control institutions. After this round of modifications, a preliminary indicator framework was developed.

There were two rounds of Delphi consultation, and we invited 41 experts within Zhuhai city (frontline emergency response workers, health emergency management personnel, and health decision-makers at the city and district levels) to participate in our research. Forty-one of them agreed to participate. To ensure a diversity of perspectives, the selection of source institutions followed the principles of hierarchical and geographic representativeness, encompassing (1) the Zhuhai Municipal Center for Disease Control and Prevention, (2) all the district-level centers for disease control and prevention in Zhuhai; and (3) selected street/township-level centers for disease control and prevention. Following the designation of these institutions, individuals from relevant departments responsible for public health emergency response and management were invited to participate. All the experts were contacted via formal email invitations. The invitation letter provided detailed information regarding the study background and the Delphi procedure. Before the formal invitation was sent, potential participants were verbally informed of the voluntary nature of participation, with explicit assurance that nonparticipation would entail no adverse consequences and that no administrative mandates were employed to compel involvement. The purpose of this recruitment process was to facilitate expert evaluation of the modules comprising the preliminary framework, thereby determining whether specific components and elements within each module warranted further development.

In the first round of consultation, we evaluated the expert authority coefficient (Cr). The Cr is determined in two dimensions, namely, the familiarity coefficient (Cs) and the judgement coefficient (Ca). The former represents the expert's familiarity with the survey, whereas the latter reflects the evidence of the expert's judgement [12,13], as shown in Table 1. The

**Table 1. Judgement basis and familiarity for consultation from experts.**

| Judgement basis (Cs) | Degree | | | Familiarity (Ca) | Value |
|---|---|---|---|---|---|
| | Great deal | Moderate | Little | | |
| Theoretical analysis | 0.30 | 0.20 | 0.10 | Very familiar | 0.90 |
| Practical experience | 0.50 | 0.40 | 0.30 | More familiar | 0.70 |
| Referring to the literature | 0.10 | 0.10 | 0.10 | Average | 0.50 |
| Intuition | 0.10 | 0.10 | 0.10 | Less familiar | 0.30 |
| | | | | Unfamiliar | 0.10 |

Cr can be calculated using the formula Cr = (Cs + Ca)/2. Usually, a Cr value higher than 0.7 is considered an acceptable indicator of reliability [14].

In addition, the experts rated the importance and feasibility of the first- and second-level indicators. According to a Likert-scale format [15], scores were divided into five levels ranging from high to low (5 = very important, 1 = very unimportant). After the data were collected in the first round, the mean importance score, feasibility score, and coefficient of variation (CV) for both the first- and second-level indicators were calculated, and the indices that did not meet the mean value of importance and feasibility assignment higher than 3.5 and whose coefficient of variation lower than 0.25 were eliminated [16]. If CV > 0.25, then the research team conducted a comprehensive analysis of importance and feasibility to determine whether to retain or delete the corresponding project [14]. Moreover, public suggestions were adopted, and at least two experts' suggestions were selected as new indicators.

Prior to the initiation of the second round of consultation, the research team provided all the experts with a structured feedback report summarizing the results of the first round. The report consisted of two components: (1) Statistical summary of item screening outcomes. The report explicitly listed the indicators that had been removed following the first round, accompanied by the rationale for their deletion—specifically, failure to satisfy the predefined retention criteria, namely, a mean importance score below 3.5 or a CV exceeding 0.25. (2) New indicators derived from expert suggestions. The research team systematically collated and thematically synthesized all the textual suggestions submitted by the experts through open-ended questions during the first round. Any content that was independently proposed by at least two experts in identical or highly similar form was incorporated as a new indicator into the second-round questionnaire. Throughout the process of organizing and presenting these expert suggestions, any information that could reveal individual expert identities—such as institutional affiliation, specific job title, or distinctive phrasing—was either removed or generalized to ensure strict adherence to the principle of anonymity required by the Delphi method.

In the second round, the revised indicators were distributed. The distribution objects consisted of the experts who provided feedback in the first round, and the evaluation content included the average importance score, feasibility score, and coefficient of variation of the first- and second-level indices. The indices that did not meet the mean value of importance and feasibility assignment higher than 3.5 and a coefficient of variation lower than 0.25 were eliminated [16].

### Decision rules for item retention and exception handling

If an indicator had a CV > 0.25 but simultaneously satisfied both of the following conditions—(1) the research team determined that the item represented a conceptually indispensable core element of the construct, and (2) the mean score was ≥ 3.5—it could be retained as an exception, accompanied by an explicit justification, to mitigate subjective bias [16].

### Testing and redesign

After completing the opinion summary and developing the system indicator framework, all of the abovementioned requirements were fed back to engineers for development. We tested and applied the usability and operability of the system,

regularly collected opinions from users, discussed any problems encountered, and then modified and improved the system on the basis of valuable information according to the needs of all parties, ultimately achieving the goal of maximizing the system's effectiveness.

## Results

We used two rounds of the Delphi method to revise the preliminary framework of the information system indicators.

### Basic information

This study distributed 41 questionnaires in two rounds of the Delphi survey and collected 31 questionnaires, for an effective response rate of 75.6%. In this study, 31 experts at the city and district levels (100%) had at least intermediate or higher professional titles or graduate degrees or above. Fifteen people (48.4%) had a postgraduate degree or above, and 7 people (22.6%) held a vice senior professional title or above. The experts consisted of frontline personnel or emergency management experts with an average of 8.6 (2–19) years of experience in public health work (see Table 1). These descriptive statistics indicate that the basic advice of the experts should be helpful. The questionnaire response rate was relatively high, with a positive coefficient of 75.6% for the two rounds of experts. The familiarity level of all the experts was 0.51, the judgement coefficient was 0.93, and the Cr was 0.72 (>0.7) [17], indicating that the expert consultation results were accurate and reliable. See Tables 2 and 3.

### Results of two rounds of expert consultation

The initial two sets of PHEISs had the same primary indicator framework (S1–S3 Tables in S1 File). In the first round of the survey, the average score for the importance of first-level indicators in the PHEIS framework was 4.31 (3.58–4.74), with a CV of 0.22 (0.14–0.39). The average feasibility score was 3.67 (3.29–3.92), with a CV of 0.29 (0.20–0.35).

Table 2. Population statistics of Delphi survey experts.

|  | Classification | Number(N) | Percentage(%) |
| --- | --- | --- | --- |
| Gender | Male | 12 | 38.7 |
|  | Female | 19 | 61.3 |
| Age (years) | 21–30 | 10 | 32.3 |
|  | 31–40 | 17 | 54.8 |
|  | 41–50 | 4 | 12.9 |
|  | 51–60 | 0 | 0 |
| Education | Undergraduate | 16 | 51.6 |
|  | Postgraduate | 14 | 45.2 |
|  | Doctor | 1 | 3.2 |
| Title | Junior | 5 | 16.1 |
|  | Intermediate | 19 | 61.3 |
|  | Deputy Senior | 6 | 19.4 |
|  | Senior | 1 | 3.2 |
| Professional field | Emergency response | 21 | 67.7 |
|  | Contingency management | 10 | 32.3 |
| Work experience (years) | 0–10 | 22 | 71.0 |
|  | 11–20 | 9 | 29.0 |
|  | 21–31 | 0 | 0 |

**Table 3. Results of two rounds of expert discussions (PHEIS at the city and district levels).**

| First-level indicator | Second-level indicator | Importance | | Feasibility | | Comprehensive | |
|---|---|---|---|---|---|---|---|
| | | Score (mean±standard deviation) | CV | Score (mean±standard deviation) | CV | Score (mean±standard deviation) | CV |
| 1. Emergency preparedness | 1.1 Emergency duty (information) | 4.58±0.85 | 0.19 | 4.45±0.81 | 0.18 | 4.52±0.64 | 0.14 |
| | 1.2 Institutional management | 4.39±0.99 | 0.23 | 4.23±0.99 | 0.23 | 4.31±0.86 | 0.20 |
| | 1.3 Team management | 4.48±0.89 | 0.20 | 4.32±1.05 | 0.24 | 4.40±0.89 | 0.20 |
| | 1.4 Material management | 4.39±1.02 | 0.23 | 3.87±1.28 | 0.33* | 4.13±0.96 | 0.23 |
| | 1.5 Knowledge materials | 4.68±0.60 | 0.13 | 3.81±1.19 | 0.31* | 4.24±0.66 | 0.15 |
| | 1.6 Form management | 4.55±0.77 | 0.17 | 4.10±1.11 | 0.27* | 4.32±0.74 | 0.17 |
| | 1.7 Scheduling management | 4.58±0.81 | 0.18 | 4.48±0.81 | 0.18 | 4.53±0.62 | 0.14 |
| 2. Monitoring and early warning | 2.1 Daily monitoring | 4.61±0.80 | 0.17 | 4.19±0.83 | 0.20 | 4.40±0.65 | 0.15 |
| | 2.2 Information monitoring | 4.65±0.66 | 0.14 | 3.94±1.09 | 0.28* | 4.29±0.75 | 0.17 |
| 3. Emergency response | 3.1 Event registration | 4.61±0.88 | 0.19 | 4.26±1.12 | 0.26* | 4.44±0.91 | 0.21 |
| | 3.2 Task management | 4.71±0.74 | 0.16 | 4.29±1.01 | 0.23 | 4.50±0.71 | 0.16 |
| | 3.3 Disposal feedback | 4.71±0.64 | 0.14 | 4.29±0.97 | 0.23 | 4.50±0.71 | 0.16 |
| | 3.4 Event correction | 4.55±0.85 | 0.19 | 4.39±0.99 | 0.23 | 4.47±0.86 | 0.19 |
| | 3.5 Sudden reporting | 4.71±0.78 | 0.17 | 4.39±0.84 | 0.19 | 4.55±0.71 | 0.16 |
| | 3.6 Event report | 4.81±0.40 | 0.08 | 4.06±1.21 | 0.30* | 4.44±0.67 | 0.15 |
| | 3.7 Investigation summary | 4.45±0.77 | 0.17 | 4.39±0.84 | 0.19 | 4.42±0.74 | 0.17 |
| | 3.8 Response termination | 4.55±0.89 | 0.20 | 4.45±0.85 | 0.19 | 4.50±0.80 | 0.18 |
| 4. Event and reserve management | 4.1 Event query and management | 4.65±0.71 | 0.15 | 3.90±1.04 | 0.27* | 4.27±0.77 | 0.18 |
| | 4.2 Knowledge data statistics and query | 4.65±0.80 | 0.17 | 3.61±1.28 | 0.36* | 4.13±0.87 | 0.21 |
| 5. System management | 5.1 Organizational structure | 4.13±1.15 | 0.28* | 3.71±1.24 | 0.34* | 3.92±0.95 | 0.24 |
| | 5.2 Module configuration | 4.19±1.14 | 0.27* | 3.65±1.20 | 0.33* | 3.92±0.92 | 0.24 |

\* Retained based on conceptual importance; see Methods section for exception criteria.

According to the calculation results, the indicator "1. Daily office" did not meet the requirements (feasibility score less than 3.50). Although the feasibility score of "5. Decision Analysis" was less than 3.50; considering its importance, this indicator was temporarily retained.

In the second round of the survey, the average score for the importance of first-level indicators in the PHEIS framework at the city and district levels was 4.51 (4.13–4.90), with a CV of 0.16 (0.06–0.26). The average feasibility score was 3.89

(3.39–4.26), with a CV of 0.25 (0.16–0.29). The average score for the importance of the first-level indicators in the PHEIS framework at the city, district, and town levels was 4.29 (3.90–4.74), with a CV of 0.25 (0.16–0.33); the average feasibility score was 3.70 (3.55–4.00), with a CV of 0.30 (0.26–0.33). While the feasibility score of "5. Reserve Management" was less than 3.50, owing to the existence of second-level indicators that had not yet been removed, the 2 first-level indicators of "event management" and "resource management" were merged.

In the PHEIS framework at the city and district levels, in the first round of consultation, the average importance score of 50 second-level indicators was 4.36 (3.58–4.84), with a CV of 0.21 (0.09–0.36), and the the average feasibility score was 3.62 (2.76–4.37), with a CV of 0.33 (0.18–0.48). A total of 26 indicators that did not meet the feasibility assignment mean higher than 3.5 were excluded, and 3 second-level indicators were deleted because of the deletion of first-level indicators; thus resulted in a total of 2 first-level indicators and 29 second-level indicators being removed. In the second round of consultation, the average importance score of the 21 second-level indicators was 4.55 (4.13–4.81), with a CV of 0.18 (0.08–0.28), and the average feasibility score was 4.13 (3.61–4.48), with a CV of 0.25 (0.18–0.36); no indicators needed to be removed.

In the PHEIS framework at the city, district, and town levels, the average importance score of 33 second-level indicators was 4.21 (3.42–4.68), with a CV of 0.26 (0.15–0.43) and the average feasibility score was 3.71 (3.00–4.13), with a CV of 0.34 (0.25–0.48). One indicator that did not meet the importance assignment mean higher than 3.5 and 11 indicators that did not meet the feasibility assignment mean higher than 3.5 were excluded. In addition, 3 second-level indicators were deleted because of the deletion of first-level indicators; this resulted in a total of 2 first-level indicators and 14 second-level indicators being removed. In the second round of consultation, the average importance score of the 19 second-level indicators was 4.35 (4.10–4.61), with a CV of 0.24 (0.18–0.31), and the average feasibility score was 3.96 (3.39–4.39), with a CV of 0.30 (0.22–0.40); this indicated that the expert opinions tended to be consistent. Two indicators that did not meet the feasibility assignment mean higher than 3.5 were excluded, and 1 first-level indicator was deleted due to the removal of second-level indicators; the 2 first-level indicators with similar functions of "event management" and "reserve management" were merged.

On the basis of two rounds of expert opinions, a city- and district-level PHEIS framework consisting of 5 first-level indicators and 21 second-level indicators was ultimately formed. The main indicators are as follows: (1) emergency preparedness: arrangement of on-duty personnel, material reserves, and management of knowledge materials; (2) monitoring and early warning: daily monitoring of infectious diseases and epidemic monitoring [18]; (3) emergency response: the entire process ranging from epidemic registration, on-site verification and disposal, incident reporting to case closure; (4) event and reserve management: management and retrieval of event information, knowledge materials, and other related content; (5) system management: setting up and assigning permissions for backend system accounts; see Table 3. In addition, city-, district-, and street/township-level PHEIS framework indicators consisting of 4 first-level indicators and 17 second-level indicators were formed; see Table 4.

## Concentration and coordination of expert opinions

In this study, we calculated the concentration and coordination of expert opinions [19] on various indicators (the average score and coefficient of variation of second-level indicators). After two rounds of expert consultation, the average score for the importance rating of the second-level indicators of PHEIS at the city and district levels included in the project increased from 4.47 (3.89 to 4.84) to 4.55 (4.13 to 4.81), and the CV decreased from 0.09–0.35 to 0.08–0.28. The comprehensive score of feasibility evaluation increased from 3.99 (3.55 to 4.37) to 4.13 (3.61 to 4.48), and the CV decreased from 0.18–0.42 to 0.18–0.36. The average comprehensive score increased from 4.23 (3.72 to 4.55)-4.34 (3.92 to 4.55), and the CV decreased from 0.10–0.36 to 0.14–0.24. See Table 3.

For the second-level indicators of the city-, district-, and town-level systems, the average importance score of the included projects decreased from 4.40 (4.03 to 4.68) to 4.38 (4.13 to 4.68), and the CV increased from 0.15–0.30 to

Table 4. Results of two rounds of expert discussions (PHEIS at the city, district, and town/street levels).

| First-level indicator | Second-level indicator | Importance | | Feasibility | | Comprehensive | |
|---|---|---|---|---|---|---|---|
| | | Score (mean±standard deviation) | CV | Score (mean±standard deviation) | CV | Score (mean±standard deviation) | CV |
| 1. Emergency preparedness | 1.1 Emergency duty (information) | 4.52±0.89 | 0.20 | 4.39±0.99 | 0.23 | 4.45±0.90 | 0.20 |
| | 1.2 Institutional management | 4.23±1.18 | 0.28* | 3.97±1.30 | 0.33* | 4.10±1.02 | 0.25 |
| | 1.3 Team management | 4.39±0.99 | 0.23 | 4.06±1.24 | 0.30* | 4.23±1.04 | 0.25 |
| | 1.4 Knowledge materials | 4.61±0.84 | 0.18 | 3.87±1.31 | 0.34* | 4.24±0.91 | 0.21 |
| | 1.5 Scheduling management | 4.29±1.10 | 0.26* | 4.03±1.17 | 0.29* | 4.16±0.97 | 0.23 |
| 2. Monitoring and early warning | 2.1 Daily monitoring | 4.32±1.14 | 0.26* | 4.00±1.13 | 0.28* | 4.16±1.06 | 0.25 |
| | 2.2 Information monitoring | 4.35±1.02 | 0.23 | 3.55±1.41 | 0.40* | 3.95±1.08 | 0.27* |
| 3. Emergency response | 3.1 Event registration | 4.29±1.10 | 0.26* | 4.19±1.05 | 0.25 | 4.24±0.98 | 0.23 |
| | 3.2 Task management | 4.39±0.99 | 0.23 | 4.10±1.16 | 0.28* | 4.24±0.96 | 0.23 |
| | 3.3 Disposal feedback | 4.45±0.99 | 0.22 | 4.06±1.12 | 0.28* | 4.26±0.98 | 0.23 |
| | 3.4 Event correction | 4.39±1.09 | 0.25 | 4.19±1.14 | 0.27* | 4.29±1.03 | 0.24 |
| | 3.5 Sudden reporting | 4.52±1.00 | 0.22 | 4.26±1.06 | 0.25 | 4.39±0.93 | 0.21 |
| | 3.6 Event report | 4.68±0.83 | 0.18 | 3.94±1.26 | 0.32* | 4.31±0.91 | 0.21 |
| | 3.7 Investigation summary | 4.32±0.98 | 0.23 | 4.19±1.05 | 0.25 | 4.26±0.96 | 0.23 |
| | 3.8 Response termination | 4.45±0.93 | 0.21 | 4.29±0.94 | 0.22 | 4.37±0.88 | 0.20 |
| 4. Event and reserve management | 4.1 Event query and management | 4.19±1.14 | 0.27* | 3.68±1.08 | 0.29* | 3.94±0.88 | 0.22 |
| | 4.2 Knowledge data statistics and query | 4.13±1.23 | 0.30* | 3.55±1.36 | 0.38* | 3.84±1.05 | 0.27* |

* Retained based on conceptual importance; see Methods section for exception criteria.

0.18–0.30. The comprehensive score of feasibility evaluation increased from 3.90 (3.50 to 4.13) to 4.02 (3.55 to 4.39), and the CV decreased from 0.25–0.40 to 0.22–0.40. The average comprehensive score increased from 4.15 (3.87 to 4.36)-4.20 (3.84 to 4.45), and the CV decreased from 0.18–0.29 to 0.20–0.27. The concentration of expert opinions was found to be relatively high; see Table 4.

The Kendall harmony coefficient and coefficient of variation were used to measure the level of coordination between expert opinions [20]. The Kendall harmony coefficient ranges from 0 to 1, with higher values indicating a higher degree of coordination. In the PHEIS framework, the results revealed that after two rounds of expert consultation, the Kendall harmony coefficient decreased from 0.140–0.218 to 0.061–0.165, p<0.01, and the difference was statistically significant; see Table 5.

## Testing and redesign

After the development was completed, we continued to improve the system while using it by collecting feedback from users. From January 1 to November 30, 2024, during the process of use and improvement, we utilized information systems to handle 536 outbreaks in Zhuhai city. The emergency system provided significant assistance for onsite disposal through functions such as reporting and registration and emergency response.

**Table 5. Coordination degrees of expert opinions in two rounds.**

| | Importance | | | Feasibility | | | Comprehensive score | | |
|---|---|---|---|---|---|---|---|---|---|
| | W | chi-square | p | W | chi-square | p | W | chi-square | p |
| The first-round PHEIS at the municipal and district levels | 0.140 | 259.943 | <0.00 | 0.218 | 405.851 | <0.00 | 0.184 | 342.563 | <0.00 |
| The first-round PHEIS at the city, district, and town street levels | 0.187 | 227.669 | <0.00 | 0.173 | 210.727 | <0.00 | 0.170 | 206.616 | <0.00 |
| Second-round PHEIS at the municipal and district levels | 0.072 | 44.915 | <0.00 | 0.134 | 82.863 | <0.00 | 0.127 | 78.518 | <0.00 |
| Second-round PHEIS at the city, district, and town street levels | 0.061 | 33.970 | 0.01 | 0.150 | 83.490 | <0.00 | 0.165 | 92.131 | <0.00 |

## Discussion

The first-level indicator with the highest expert ratings is "emergency response", with comprehensive scores of 4.58 and 4.37 for the city/district-level system and the city/district/town-level system, respectively. Emergency response includes the entire process of investigating infectious disease outbreaks, and through information technology, the efficiency of onsite response by emergency team members has improved. This module has the highest utilization rate in daily use; therefore, its importance and operability are recognized by experts.

The comprehensive scores of "emergency preparedness" in the two systems are 4.34 and 4.16, making it the second most commonly used module. Emergency preparedness is the foundation of health emergency work, including the construction of personnel teams, the reserve of various emergency materials, the preparation of knowledge materials, the development of emergency plans, and holding regular training and drills on a daily basis. Only when the abovementioned preparations are fully implemented can resources be immediately mobilized for disposal in the event of an epidemic.

"Monitoring and early warning" is the third most commonly used module in the system. This function is reflected mainly to the collection of information related to sudden public health emergencies through multiple channels, such as hospitals' online reporting of legally recognized infectious diseases, school doctors' reporting of student symptom monitoring data, media information monitoring, etc., to discover relevant information that may halt infectious disease outbreaks in a timely manner and intervene in the early disposal of the epidemic.

In the second round of expert consultation, the mean comprehensive scores increased, and the CVs decreased; however, the Kendall coefficient of concordance (W) decreased from 0.140–0.218 in the first round to 0.061–0.165 in the second round. This phenomenon may be explained by the following two considerations: (1) Indicator screening and reduced discrimination. The first-round consultation included 50 and 33 second-level indicators, respectively, and expert ratings exhibited considerable variability across items, enabling Kendall's W to effectively capture the consistency of rank ordering among panel members. Following the first-round screening, only 21 and 17 highly consistent second-level indicators were retained in the second round, and the experts assigned high scores of 4 or 5 to the majority of these items. When nearly all ratings are concentrated within a narrow high-score band, the discriminatory capacity of Kendall's W—a nonparametric statistic designed to measure rank-order agreement—is substantially diminished. This decline thus reflects the high degree of consensus already achieved on the retained items rather than an increase in expert disagreement. (2) Expert attrition and panel heterogeneity. Ten experts withdrew during the second round of consultation (attrition rate: 24.4%). Such attrition may have altered the composition of the panel. Heterogeneity among experts in terms of professional background or institutional affiliation may also have influenced the W statistic. In summary, the observed decrease in Kendall's W in the present study is more attributable to the statistical characteristics of reduced discrimination following item screening and the inherent nature of the rating data than to a substantial deterioration in the degree of expert coordination. Several published Delphi studies have reported similar phenomena [20,21], and Kendall's W values ranging from 0.1–0.3 with P < 0.05 are accepted in the methodological literature as indicating an acceptable level of concordance.

Parts of the preliminary framework developed in this study were ultimately deleted, i.e., risk assessment [22], emergency training, decision analysis, and daily office work; this was mainly due to the low daily usage rate of these modules,

their incomplete functional development, and experts' belief in their weak operability and low expectations for module improvement. However, these features still need to be explored in the future. Some experts argued that there is a high turnover of personnel in grassroots disease control centres and that information systems should be streamlined as much as possible; otherwise, excessive training time costs may result, which is not conducive to the promotion and popularization of the system.

In the "emergency response" module, we aim to modify the system with the goal of convenience, making it user friendly and easy to use [23] and avoiding making fundamental changes to existing work procedures, thereby stimulating the enthusiasm of frontline public health personnel and emergency management personnel to use the system. In terms of investigation reports, intelligent report generation is currently widely used in medical imaging [24]. This study also achieved the intelligent generation of survey reports through web crawlers (which automatically capture key fields of survey data), greatly reducing the workload of report writing. In terms of food poisoning investigations, a QR code scanning [25] function has been developed for individual cases. Cases can scan the code on their own mobile phones and enter information under the guidance of the investigation personnel, thereby accelerating the investigation process.

At present, decision analysis functions are relatively lacking. Owing to the difficulty of achieving qualitative and quantitative decision indicators in PHEISs, decision-making still requires experts to combine experience to make decisions. In addition, the lack of timely and accurate onsite data collection during the use of the PHEIS resulted in a lack of effective data support for this module, making it unable to function properly. The original plan of this study was to apply commonly used data analysis models for decision-makers, such as ARIMA product seasonal models [26] and machine learning techniques [27], in daily monitoring and early warning, pathogen prediction, and epidemic evolution prediction. However, owing to the low usage rate of this module and the low recognition level of experts towards it, after two rounds of consultation, the first- and second-level indicators related to this function were deleted.

## Advantages and limitations

This study has the following advantages. First, it involves public health professionals and engineers in the development process of information systems, thereby reducing the possibility of significant changes after system construction, saving system development costs, and improving system availability. Second, it uses the Delphi method, which overcomes the shortcomings of the expert experience method, such as an indicator selection that is too arbitrary, a lack of brainstorming [9], or too much emphasis on the use of user interests by development units. However, this method also has the disadvantage of being labour intensive, requiring health professionals to fully participate in the development of the information system within one year. Third, the integration and data sharing of multiple systems have been completed, integrating different data into the public health emergency information system to enhance monitoring and early warning functions [18,28]. Based on the preliminary investigation, databases with which the PHEIS plans to connect include the following: 1. epidemic warning information, including basic information about schools, suspected case information, etc., which is provided by the school symptom monitoring system [29], with the aim of decreasing the need for public health personnel to conduct repeated onsite investigations to obtain basic information about individual cases; 2. laboratory system information, in which sample testing information is generated from onsite case investigation information, saving the input operation of sample delivery forms; subsequently, such information will be obtained by capturing cases and collecting sample testing information from other sites, which means that cumbersome steps such as reporting results and printing and archiving test reports can be omitted.

In this study, mean scores and the CVs were used as the consensus criteria [16,18]. Although this approach has precedents in related fields, median-based measures, interquartile range (IQR), and percentage agreement are more commonly recommended in the Delphi methodology literature. Future studies may consider reporting multiple consensus metrics concurrently to increase the robustness of the findings.

Although indicator selection was guided primarily by prespecified statistical thresholds (CV ≤ 0.25 and mean score ≥3.5), a limited number of indicators were retained despite exceeding the CV threshold on the basis of their conceptual

importance and content validity. This introduces a degree of subjectivity into the selection process. Nevertheless, in Delphi studies where rigid adherence to statistical cut-offs would result in the exclusion of conceptually indispensable items, exception-based retention—particularly when cross-validated against other quantitative metrics—remains a methodologically acceptable compromise. To mitigate this limitation, all exceptions were documented with explicit justification and were reviewed and confirmed by two independent members of the research team.

This study was conducted mainly in Zhuhai, Guangdong Province. However, Zhuhai is a developed coastal area in China, and its common types of infectious disease outbreaks may differ from those of other regions in the country, which limits the applicability and generalizability of the current findings. Owing to the recent establishment of district-level and street/township-level disease control centres in Zhuhai, some experts invited to participate in this Delphi study did not possess extensive work experience.

## Conclusion

This study used the Delphi method to scientifically construct a set of PHEISs at the city, district, and town levels in Zhuhai, which can greatly improve emergency responses, data collection, and command decision-making for infectious disease outbreaks. This is the first study to focus on the construction of an information system for infectious disease outbreaks, and its construction method, process, and results have significant implications for the construction of PHEISs in various provinces and cities in China. This study elaborates on the PHEIS construction process and combines it with the Delphi method to compensate for the shortcomings of the expert experience method. Owing to the short usage time of the newly constructed system, we analyzed the data during the usage process on the basis of the system's usage situation, further demonstrating and improving the practicality of the system.

On the basis of the findings of this Delphi consensus study, we propose the following actionable recommendations:

1. Capacity building at the grassroots level. Given the recent establishment of street/township-level centers for disease control and prevention in Zhuhai and the anticipated nationwide expansion of similar institutions, targeted training programs should be developed to ensure that staff at all administrative levels—municipal, district, and street/township—possess the requisite skills to operate the system effectively. Training curricula should emphasize streamlined workflows to minimize the onboarding burden in settings with high staff turnover.

2.  Periodic evaluation and iterative refinement. The indicator framework should be subjected to regular review cycles to accommodate evolving public health threats, emerging infectious diseases, and feedback from end-users. The high retention rates and favorable consensus scores observed in the "emergency preparedness" and "emergency response" modules in this study underscore their operational importance among frontline public health professionals and their central role in the daily functioning of the system. As system maturity and data infrastructure improve, indicators excluded during the current consensus process due to feasibility concerns should be re-evaluated.

3. Integration with existing health information systems. To maximize the utility of the public health emergency information system and reduce redundant data entry, we strongly recommend that the framework be integrated with existing surveillance platforms, including hospital-based notifiable disease reporting systems, school symptom monitoring networks, and laboratory information management systems. Such integration will enhance the timeliness and completeness of data captured within the public health emergency information system and support more robust monitoring and early warning functions.

The framework presented herein offers a pragmatic, evidence-based pathway toward a more resilient and responsive public health emergency management system. Further research is warranted to evaluate the real-world impact of this PHEIS framework on key performance metrics, including outbreak response times, data completeness, and decision-making efficiency. Additionally, validation studies conducted in other geographic regions would help establish the generalizability and adaptability of the framework beyond the specific conditions of Zhuhai city.

 

## Supporting information

**S1 File. Results of the first round of Delphi consultation. S1 Table.** First-level indicators: Importance scores. **S2 Table**. Second-level indicators (city and district systems): Importance scores. **S3 Table.** Second-level indicators (city, district, and town/street-level systems): Importance scores. **S4 Table.** First-level indicators: Feasibility scores. **S5 Table.** Second-level indicators (city and district systems): Feasibility scores. **S6 Table.** Second-level indicators (city, district, and town/street-level systems): Feasibility scores. **S7 Table.** Second-level indicators (city and district systems): Comprehensive scores. **S8 Table.** Second-level indicators (city, district, and town/street-level systems): Comprehensive scores.
(XLSX)

**S2 File. Results of the second round of Delphi consultation. S9 Table.** First-level indicators (city and district systems): Importance scores. **S10 Table.** First-level indicators (city, district, and town/street-level systems): Importance scores. **S11 Table.** Second-level indicators (city and district systems): Importance scores. **S12 Table.** Second-level indicators (city, district, and town/street-level systems): Importance scores. **S13 Table.** First-level indicators (city and district systems): Feasibility scores. **S14 Table.** First-level indicators (city, district, and town/street-level systems): Feasibility scores. **S15 Table.** Second-level indicators (city and district systems): Feasibility scores. **S16 Table.** Second-level indicators (city, district, and town/street-level systems): Feasibility scores. **S17 Table.** Second-level indicators (city and district systems): Comprehensive scores. **S18 Table.** Second-level indicators (city, district, and town/street-level systems): Comprehensive scores.
(XLSX)

## Acknowledgments

The authors wish to extend their sincere gratitude to the study participants for their valuable contributions to this research. Appreciation is also expressed to both current and former researchers and staff members. Special thanks are due to Feng Ruan, Xuebao Zhang, Long Chen, Xihe Ni, Ying Fan, Chaomin Zhao, and Junfeng Xiao for their assistance with the research and administrative support.

## Author contributions

**Funding acquisition:** Sicheng Huang.

**Investigation:** Sicheng Huang, Xuebao Zhang, Long Chen, Xihe Ni, Ying Fan, Chaomin Zhao.

**Methodology:** Sicheng Huang.

**Project administration:** Sicheng Huang.

**Resources:** Sicheng Huang, Xuebao Zhang.

**Software:** Sicheng Huang, Junfeng Xiao.

**Writing – original draft:** Sicheng Huang.

**Writing – review & editing:** Sicheng Huang, Feng Ruan.

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
