## [Decision Letter · Decision Letter 0]

8 Apr 2026

PONE-D-26-01579Construction of a Public Health Emergency Information System Framework: A Case Study of Zhuhai City, ChinaPLOS One

Dear Dr. Ruan,

Thank you for submitting your manuscript to PLOS ONE. After careful consideration, we feel that it has merit but does not fully meet PLOS ONE’s publication criteria as it currently stands. Therefore, we invite you to submit a revised version of the manuscript that addresses the points raised during the review process.

We look forward to receiving your revised manuscript.

Kind regards,

Pasyodun Koralage Buddhika Mahesh

Academic Editor

PLOS One

Journal Requirements:

6. We note that there is identifying data in the Supporting Information file <Supplementary material.zip>. Due to the inclusion of these potentially identifying data, we have removed this file from your file inventory. Prior to sharing human research participant data, authors should consult with an ethics committee to ensure data are shared in accordance with participant consent and all applicable local laws.

-Location data

7. We are unable to open your Supporting Information file [Supplementary material.zip]. Please kindly revise as necessary and re-upload.

Reviewers' comments:

Reviewer's Responses to Questions

**Comments to the Author**

1. Is the manuscript technically sound, and do the data support the conclusions?

Reviewer #1: Partly

Reviewer #2: Yes

2. Has the statistical analysis been performed appropriately and rigorously? 

Reviewer #1: Yes

Reviewer #2: Yes

3. Have the authors made all data underlying the findings in their manuscript fully available?

Reviewer #1: Yes

Reviewer #2: Yes

4. Is the manuscript presented in an intelligible fashion and written in standard English?

Reviewer #1: Yes

Reviewer #2: Yes

5. Review Comments to the Author

Reviewer #1: This manuscript addresses an important and relevant topic: the development of a public health emergency information system (PHEIS) framework using a Delphi approach. The multi-level design and incorporation of real-world application are notable strengths, and the study has potential practical value for public health systems.

However, several methodological and reporting concerns limit confidence in the findings. Addressing the issues below will substantially improve the rigor, transparency, and interpretability of the study.

Major Comments

1. Lack of clarity on expert selection and recruitment

Comment:

Although expert characteristics are presented (Table 2), the manuscript does not clearly describe the criteria used to select experts, nor the sampling strategy or recruitment process. This limits transparency and raises concerns about the validity and representativeness of the Delphi panel.

Action required by authors:

• Clearly define eligibility criteria for experts (e.g., qualifications, years of experience, roles in public health emergency systems).

• Specify the sampling strategy (e.g., purposive, convenience, nomination-based).

• Describe the recruitment process (e.g., email invitation, institutional selection, voluntary participation).

• Indicate whether any screening or inclusion/exclusion criteria were applied.

• Add these details to the Methods section (Delphi method subsection).

• Acknowledge potential selection bias and limited generalizability in the Discussion under limitations.

2. Inadequate justification for consensus criteria

Comment:

The study uses mean scores (≥3.5) and coefficient of variation (<0.25) as thresholds for indicator retention. These cut-offs are not justified, and there are no universally accepted standards supporting this approach. In Delphi studies, consensus is more commonly assessed using median-based measures, interquartile ranges (IQR), or percentage agreement.

Action required by authors:

• Provide explicit justification with appropriate references for using mean and CV thresholds; OR

• Revise the analysis to include more widely accepted Delphi consensus measures, such as:

o Median and IQR

o Percentage agreement among experts

• If retaining the current method:

o Clearly justify why mean and CV were selected over standard approaches

o Discuss the limitations of this method in the Discussion section

• Ensure the chosen method aligns, where possible, with established reporting standards such as CREDES guideline.

3. Insufficient description of the Delphi process

Comment:

The Delphi methodology is insufficiently described, particularly regarding the iterative process between rounds and the rationale for conducting only two rounds. This affects reproducibility and methodological transparency.

Action required by authors:

• Expand the Methods section to include:

o How feedback was provided between rounds (e.g., summary statistics, anonymized expert comments)

o Whether participants were allowed to revise their responses

o The mode of administration (e.g., online survey, email-based; specify if this was an e-Delphi approach)

• Provide a clear justification for limiting the process to two rounds (e.g., consensus achieved, response stability, feasibility constraints).

• Clarify whether this was a classical Delphi, modified Delphi, or e-Delphi approach.

4. Inconsistency in applying predefined inclusion criteria

Comment:

There appears to be inconsistency in applying the predefined thresholds, as some retained indicators exceed the stated coefficient of variation criteria (Tables 3 and 4). This undermines methodological consistency.

Action required by authors:

• Reassess all indicators in Tables 3 and 4 to ensure alignment with predefined inclusion criteria.

• Either:

o Apply the criteria consistently across all indicators, OR

o Clearly explain exceptions and decision-making rationale (e.g., conceptual importance overriding statistical thresholds).

• If exceptions are retained:

o Explicitly state this in the Methods as part of the decision rules

o Acknowledge this as a source of subjectivity in the Discussion under limitations.

• Ensure full consistency between Methods and Results sections.

5. Misinterpretation of Kendall’s coefficient of concordance (W)

Comment:

Kendall’s W decreases across rounds (Table 5), indicating reduced agreement among experts. However, the manuscript interprets this as an improvement in consensus. This interpretation is inconsistent with standard Delphi methodology, where agreement is expected to increase with successive rounds.

Action required by authors:

• Correct the interpretation of Kendall’s W in both the Results and Discussion sections.

• Clearly state that a decrease in W reflects reduced agreement, not improved consensus.

• Provide a plausible explanation for this finding (e.g., indicator reduction, panel heterogeneity, attrition between rounds).

• Reframe conclusions regarding consensus accordingly.

• Explicitly acknowledge this issue in the Discussion under limitations.

6. Overall methodological alignment with reporting standards

Comment:

Overall, substantial revisions are required to improve methodological transparency and align the study with accepted standards such as the CREDES guideline.

Action required by authors:

• Revise the manuscript to ensure alignment with established Delphi reporting standards, particularly CREDES.

• Ensure comprehensive reporting of:

o Expert selection and characteristics

o Consensus definition and justification

o Iterative feedback process

o Stability and agreement across rounds

• Consider adding a statement in the Methods section indicating adherence (or partial adherence) to CREDES.

• Improve overall clarity, transparency, and reproducibility of the methodology.

Reviewer #2: Congratulations on your valuable attempt in constructing a public health information system using the Delphi technique! This is a timely and much-needed contribution towards ensuring timely responses during public health emergencies.

Please find some comments for further improvement of this manuscript mentioned below:

Ethics statement (page 4 of the PDF document)

• In the ethics statement, it is stated that “Verbal assent was obtained from all participants prior to the study”. It is suggested to clarify what this means, as assent is usually taken from minors/children.

Abstract

• It is suggested to separate the Background from the objectives in the abstract. Please mention the objective clearly.

• Please be mindful of the tense. Some sentences were written in the present tense

Manuscript

• It is good if the initial indicator framework could be shown (page 5, line 8)

• Please avoid starting sentences with abbreviations (Cr) (page 6, line 1)

• Please provide references/ rationale for your cut-offs (page 6- line 2, page 7- lines 2&3)

• It is suggested to move the sentence on response rate (line 20, page 7) to the beginning of the Basic information section, before describing sample characteristics.

Conclusion

• The ending seemed a bit abrupt. Please end the manuscript with a strong recommendation.

Thank you

6. PLOS authors have the option to publish the peer review history of their article (what does this mean?). If published, this will include your full peer review and any attached files.

Reviewer #1: **Yes:** I.O.K.K.Nanayakkara

Reviewer #2: No

---

## [Author Response · Author response to Decision Letter 1]

27 Apr 2026

We have carefully revised the manuscript in accordance with the comments and suggestions provided by the editor and both reviewers. Detailed point-by-point responses to all comments are provided in the accompanying document titled "Response to Reviewers."

---

## [Decision Letter · Decision Letter 1]

21 May 2026

Construction of a Public Health Emergency Information System Framework: A Case Study of Zhuhai City, China

PONE-D-26-01579R1

Dear Dr. Ruan,

We’re pleased to inform you that your manuscript has been judged scientifically suitable for publication and will be formally accepted for publication once it meets all outstanding technical requirements.

Kind regards,

Pasyodun Koralage Buddhika Mahesh

Academic Editor

PLOS One

Additional Editor Comments (optional):

Reviewers' comments:

Reviewer's Responses to Questions

**Comments to the Author**

1. If the authors have adequately addressed your comments raised in a previous round of review and you feel that this manuscript is now acceptable for publication, you may indicate that here to bypass the “Comments to the Author” section, enter your conflict of interest statement in the “Confidential to Editor” section, and submit your "Accept" recommendation.

Reviewer #1: All comments have been addressed

Reviewer #2: All comments have been addressed

2. Is the manuscript technically sound, and do the data support the conclusions?

Reviewer #1: Yes

Reviewer #2: (No Response)

3. Has the statistical analysis been performed appropriately and rigorously? 

Reviewer #1: Yes

Reviewer #2: (No Response)

4. Have the authors made all data underlying the findings in their manuscript fully available?

Reviewer #1: Yes

Reviewer #2: (No Response)

5. Is the manuscript presented in an intelligible fashion and written in standard English?

Reviewer #1: Yes

Reviewer #2: (No Response)

6. Review Comments to the Author

Reviewer #1: The authors have substantially addressed the previous comments. The revised manuscript now provides clearer details on expert selection, recruitment, the modified e-Delphi process, feedback between rounds, decision rules for indicator retention, and alignment with CREDES guidance. The issue regarding consensus criteria has been reasonably addressed through justification and discussion of limitations. The interpretation of Kendall’s W has also been improved.Overall, I consider the manuscript much improved and suitable for acceptance after minor editorial revision.

Reviewer #2: Dear authors,

I would like to congratulate you all once again! Thank you for taking my comments into consideration.

7. PLOS authors have the option to publish the peer review history of their article (what does this mean?). If published, this will include your full peer review and any attached files.

Reviewer #1: **Yes:** I.O.K.K.Nanayakkara

Reviewer #2: **Yes:** Herath Mudiyanselage Chathurika Dulmini Herath

---

## [Editor Report · Acceptance letter]

PONE-D-26-01579R1

PLOS One

Dear Dr. Ruan,

I'm pleased to inform you that your manuscript has been deemed suitable for publication in PLOS One. Congratulations! Your manuscript is now being handed over to our production team.

Kind regards,

on behalf of

Dr. Pasyodun Koralage Buddhika Mahesh

Academic Editor

PLOS One